# Fast Fault Diagnosis in Industrial Embedded Systems Based on Compressed Sensing and Deep Kernel Extreme Learning Machines

**DOI:** 10.3390/s22113997

**Published:** 2022-05-25

**Authors:** Nanliang Shan, Xinghua Xu, Xianqiang Bao, Shaohua Qiu

**Affiliations:** National Key Laboratory of Science and Technology on Vessel Integrated Power System, Naval University of Engineering, Wuhan 430033, China; nanliang@stu.xmu.edu.cn (N.S.); xinghuaxv@163.com (X.X.); baoxianqiang@nudt.edu.cn (X.B.)

**Keywords:** fast fault diagnosis, industrial embedded systems, compressed sensing, deep kernel extreme learning machine

## Abstract

With the complexity and refinement of industrial systems, fast fault diagnosis is crucial to ensuring the stable operation of industrial equipment. The main limitation of the current fault diagnosis methods is the lack of real-time performance in resource-constrained industrial embedded systems. Rapid online detection can help deal with equipment failures in time to prevent equipment damage. Inspired by the ideas of compressed sensing (CS) and deep extreme learning machines (DELM), a data-driven general method is proposed for fast fault diagnosis. The method contains two modules: data sampling and fast fault diagnosis. The data sampling module non-linearly projects the intensive raw monitoring data into low-dimensional sampling space, which effectively reduces the pressure of transmission, storage and calculation. The fast fault diagnosis module introduces the kernel function into DELM to accommodate sparse signals and then digs into the inner connection between the compressed sampled signal and the fault types to achieve fast fault diagnosis. This work takes full advantage of the sparsity of the signal to enable fast fault diagnosis online. It is a general method in industrial embedded systems under data-driven conditions. The results on the CWRU dataset and real platforms show that our method not only has a significant speed advantage but also maintains a high accuracy, which verifies the practical application value in industrial embedded systems.

## 1. Introduction

With the development of modern industrial systems and the pursuit of extreme efficiency, complex industrial systems are gradually automated, sophisticated and integrated [1]. Mechanical failure of key components can easily lead to the collapse of the entire system. To accurately capture the internal status information of key components, high-precision industrial sensors are used to obtain time-series monitoring signals for health status assessment. Due to a large number of checkpoints, high sampling rate and long detection time, the health monitoring system has acquired massive status data. On the one hand, it has prompted the fault diagnosis of complex industrial systems to move into the “data-driven” era [2], on the other hand, it has also brought great difficulty to fast fault diagnosis in resource-constrained industrial embedded systems [3].

The existing research on fast fault diagnosis of the rotating components has achieved valuable results. Such as a data-driven bearing prognostic scheme that was designed based on novel health indicators and gated recurrent unit network [4]. The work [5] studies the statistical characteristics of the spalling propagation for rotating components, which is conducive to predicting the occurrence of failures. The work [6] enables dynamic modeling of defect extension and appearance of rotating components, which helps us to predict service life based on defect models. However, there is still a lot of work to be done to achieve fast fault diagnosis in resource-constrained industrial embedded systems. For resource-constrained industrial embedded systems, we believe that we should first achieve lightweight data and optimize the data sampling process. Data sampling refers to monitoring the health status of the target device for a long time through a large number of sensors. The optimization of data sampling mainly adopts data compression. The sampling methods still follow the Nyquist sampling theorem [7], and the sampled signals contain a lot of redundant information. For fault diagnosis, we pay more attention to the accuracy and timeliness in implementing fault diagnosis in embedded systems, so we optimize in terms of feature extraction and fault classification. Feature extraction refers to obtaining data features from a signal through signal processing methods. The optimization in feature extraction mainly adopts transform domain analysis, which includes principal component analysis [8] and discrete cosine transform [9]. Fault classification refers to predicting the fault category of unknown signals by learning data features. The research on fault classification optimization is relatively mature and includes the latest algorithms: ANN [10], SPBO-SDAE [11], PSO-DNN [12] and CS-IMSNs [13].

Starting from the above three aspects, this work further explores the three limitations:1.The original sampling signals contain a lot of redundant information, which will greatly increase the pressure for subsequent data transmission, storage and calculation.2.Features extracted from the original sampling signals have problems such as high computational complexity, long time consumption and incomplete reflection of fault features.3.Existing fault diagnosis methods cannot effectively balance efficiency and accuracy in resource-constrained environments and cannot effectively classify sparse signals.

Recently, industrial embedded systems [14] have drawn significant attention when applied to industrial process control [15] and monitoring [16,17,18] and edge intelligent systems [19,20]. Unlike the loose requirements found in traditional Industrial IoT (IIoT) applications, resource-constrained industrial embedded systems set stricter requirements for low latency and small memory. The existing lightweight methods are mainly reflected in model optimization, including network pruning, weight quantization and knowledge distillation. However, there is no method designed for the whole process of fault diagnosis for data sampling, feature extraction and fault classification.

Compressing the data from the data sampling source will significantly reduce the subsequent computational complexity. Compressed sensing (CS) theory, which integrates data sampling and compression, may give us some inspiration [21]. CS adopts an approximate optimal sampling scheme, theoretically obtains all the information contained in the original signal and effectively realizes the dimension reduction of the signal. It frees data sampling from the Nyquist theory and can achieve signal compression through nonlinear projection in the transformation domain, thereby reducing the mass of redundant data sampling into a small amount of useful information sampling. Therefore, the storage and calculation costs required for feature extraction and fault classification are greatly reduced. Integrating data sampling and data compression, the efficient information perception method provides a brand-new idea for data sampling and processing in real-time monitoring systems.

Some studies have introduced the concept of CS [22,23,24], and the general process of which includes compressed sampling, data transmission and compressed signal reconstruction. Then, the reconstructed signal will be input to the feature extraction and fault classification model. Although these methods reduce the requirements for data storage and data transmission, the subsequent operations are still based on the reconstructed time-domain signals. Therefore, there is still significant room for improvement in diagnosis time if the sparse compressed sampling signal can be directly used in fault diagnosis [25]. Since the compressed sampling signals contain the information of fault characteristics in a small amount of data, as long as a suitable fault diagnosis model is constructed, efficient fault diagnosis can be achieved.

Deep Extreme Learning Machine (DELM) [26] is a multi-layer neural network that uses Extreme Learning Machine-AutoEncoder (ELM-AE) as the basic unit of unsupervised learning, which performs greedy layer-by-layer unsupervised training on input data. It is widely used because it is faster than other deep neural networks in terms of model training and weight update of hidden layers [27]. Compared with the ELM-AE, which has a single hidden layer, the DELM can extract more complex fault features in the low-dimensional observation signal. Therefore, the DELM is an ideal fault classification model that integrates the accuracy of DNN and the speed of traditional machine learning [28]. In order to make more accurate fault classifications for the sparse signal after compressed sampling, this paper introduces the kernel function into DELM to obtain the fault diagnosis module DKELM. DKELM maps low-dimensional nonlinear inseparable data features to high-dimensional space to make it linearly separable, so it has better adaptability to sparse signals.

Combining the advantages of CS and DKELM, this paper proposes a general diagnosis method to realize the functional integration of signal compressed sampling, feature adaptive extraction and fast fault diagnosis. Meanwhile, the Particle Swarm Optimization (PSO) Algorithm [29] is used to optimize the nodes of each hidden layer, the regularization coefficient, the penalty coefficient and the core parameters of the DKELM so as to achieve the best fault classification results. The main contributions of this paper are summarized as follows:1.A data-driven general method for fast fault diagnosis is proposed. This method only needs a small amount of compressed sampling data to achieve fast and high-accuracy fault diagnosis, which dramatically reduces diagnosis time in industrial embedded systems.2.A new adaptive feature extraction method for high-frequency monitoring signals is proposed. Our research found that not all IIoT monitoring signals are strictly sparse, a sparse basis should be added before compressed sampling, and then the transformed high-dimensional signal should be compressed by the observation matrix. This can be seen as a feature representation.3.An improved DKELM fault diagnosis module is constructed. The module can better adapt to sparse signals and provides a new idea for fast fault diagnosis in industrial embedded systems.

The remainder of this paper is organized as follows. Section 2 introduces a fast fault diagnosis method based on CS-DKELM. Section 3 verifies the effectiveness of the method. In addition, our method is compared with existing methods, and the effects of important factors are analyzed. Finally, Section 4 draws the conclusions.

## 2. Proposed CS-DKELM Method

Aiming at the limitations of current fault diagnosis methods, such as low computational efficiency, difficulty in feature extraction, the poor balance between efficiency and accuracy and insensitivity to sparse signals, a general method for data sampling and fast fault diagnosis based on CS and DKELM was proposed. This method has two main advantages: one is that it is essentially a classic machine learning algorithm, so its model and computational complexity are lighter than deep learning algorithms, which is more suitable for deployment in industrial embedded systems; the second is that it is especially improved for the sparse signal after compressed sampling, which not only shortens the diagnosis time but also maintains high accuracy. It includes a CS-based data sampling module, which achieves substantial compression on the premise of retaining fault information, greatly reducing the computational complexity of subsequent feature extraction and fault classification. It also contains a DKELM-based fast fault diagnosis module. Through multi-layer nonlinear learning of low-dimensional sampled values, adaptive extraction of fault features and fast diagnosis of sparse signals are realized. The method in this paper is a general method for fast fault diagnosis, which is suitable for high-frequency monitoring signals in industrial systems. The vibration signal of rotating machinery is used here as a demonstration. The procedure of the proposed CS-DKELM method is shown in Figure 1.

### 2.1. Data Processing

The experimental data were provided by the public bearing data set of Case Western Reserve University Bearing Data Center [30]. This dataset is the drive-end vibration data sampled by a SKF6205-2RS deep groove ball bearing at a sampling frequency of 48 kHz and a load of 0 HP. It includes four health conditions: Normal status (N), Outer race fault (O), Inner race fault (I) and Ball fault (B). Among them, each fault is divided into three categories according to the damage diameter, which are 0.07, 0.14 and 0.21 mm, respectively. Therefore, there are 10 kinds of sample data in total. We set the sliding window size to 4800 points for data processing. A total of 100 samples of each type of fault were collected. We marked the preprocessed data set as *D*, the dimension of which is 1000 × 4800. We randomly selected 70% of the data as training data and the remaining 30% as test data. Then, the preprocessed data set *D* was compressed, sampled and normalized. The measurement matrix was a Gaussian random matrix, and the compressed rate was CR=80%. The compressed sampling data set $D^{\prime }$ was obtained, with a dimension of 1000 × 960. In subsequent experiments, the data set $D^{\prime }$ will be used as the input data for fast fault diagnosis. The details are shown in Table 1.

The mechanical vibration signal is a high-frequency periodic signal. Calculating the monitoring signals directly will exceed the maximum length preset by the computer array in the industrial embedded system, resulting in memory overflow. Therefore, the data must be preprocessed into data samples. This method uses the sliding window for the monitoring signal and integrates the time series data into a data set $D={\left\{d^{i},l^{i}\right\}}_{i=1}^{M}$, where $d^{i}$ is the preprocessed monitoring signals, $l^{i}$ is the corresponding label and *M* is the total number of samples. The data sample set can be randomly selected and divided into a training set $D_{train}$ and a test set $D_{test}$.

### 2.2. CS-Based Compressed Sampling

In view of the large scale of the monitoring signals in industrial embedded systems, it is inconvenient to transmit, store and calculate. Donoho [21] proposed a Compressed Sensing (CS) theory based on signal sparsity, which broke the traditional lossless signal restoration theory based on signal bandwidth as a priori (Nyquist theory). The principle of CS is to take advantage of the sparsity of the signal in the transform domain. A low-dimensional compressed sampled signal can be obtained by projecting the original signal with a measurement matrix that is incoherent to the transform domain. The original signal can be restored with high fidelity by combinatorial optimization methods. Combining compression and sampling greatly reduces the amount of sampling of the original signal while maintaining signal integrity. The compressed sampling signal can be expressed as follows:(1)y=Φx=ΦΨs.
where $\Phi \in R^{M\times N}\left(M\ll N\right)$ is the measurement matrix. $x\in R^{N}$ is the original signal. Ψ is the dictionary matrix or sparse basis. s is the sparse factor, which contains a few non-zero values.

In the process of CS, the Compressed Rate (CR) reflects the degree of compression of the original signal, which is defined as follows
(2)$$\mathrm{CR}=\frac{N-M}{N}\times 100\%.$$
where N is the dimension of the original signal. M is the dimension of the compressed signal. By adjusting the CR, different degrees of compression can be achieved. The larger the CR is, the less compressed sampling data can be obtained.

There are two prerequisites for the realization of CS. One is that the original signal must meet the sparseness in a certain transform domain [21], and the other is that the measurement matrix must meet the restricted isometric property (RIP) [31]. The compressed sampling data that meets these two conditions can be accurately reconstructed. Candes E [32] proved that the Random Gaussian Matrix could satisfy RIP with high probability and has good universality, so it is widely used in CS. This feature makes it possible to generate a Random Gaussian Matrix of appropriate size as a measurement matrix to compress and sample the original signal when the signal is sparse in the unknown space.

Our research found that the high-frequency periodic monitoring signals are not strictly sparse in the transform domain, so they cannot be directly compressed via sampling. This module adds a Fourier transform matrix before compressed sampling to transform the time-domain signal into the frequency-domain (as shown in Figure 2a), and then uses an observation matrix (Gaussian random matrix) that is not incoherent to the sparse matrix (Discrete cosine transform matrix) to project the transformed frequency-domain signal onto a low-dimensional space to achieve compressed sampling. The obtained signal is a frequency-domain compressed sampling signal (as shown in Figure 2b), which contains almost all the fault information of the original signal and can be directly used in the subsequent fault diagnosis process.

### 2.3. DKELM-Based Fast Fault Diagnosis

An Extreme Learning Machine (ELM) [33] is proposed for the training of a single hidden layer feedforward neural network. It is very different from the traditional BP network. It does not need to solve iteratively but adopts a batch-processing “one-step” training method. We randomly selected the weights and biases of hidden nodes and completed the training by the inverse operation of the output weights. It greatly improves the generalization ability and training speed of the network and also reduces the amount of calculation and search space.

Giving *N* training samples
(3)$$Z={\left\{x_{i},t_{i}\right\}}_{i=1}^{N},x_{i}\in R^{n},t_{i}\in R^{m}.$$

Then the ELM model with nodes can be expressed as
(4)$$\sum_{j=1}^{L}\beta_{j}g\left(\omega_{j}x_{i}+b_{j}\right)=t_{i}.$$
where *L* is the hidden layer nodes. $\omega_{j}={\left[\omega_{j1},\omega_{j2},\cdots ,\omega_{jn}\right]}^{T}$ is the weight vector connecting the jth hidden layer node and the input nodes. $\beta_{j}={\left[\beta_{j1},\beta_{j2},\cdots ,\beta_{jm}\right]}^{T}$ is the weight vector connecting the jth hidden layer node and the output nodes. g() is the activation function. The above formula can be abbreviated as
(5)Hβ=T.
where H is N×L hidden layer output matrix. $\beta ={\left[\beta_{1},\beta_{2},\cdots ,\beta_{L}\right]}_{L\times m}^{T}$ is the output weight matrix. $T={\left[t_{1},t_{2},\cdots ,t_{n}\right]}_{N\times m}^{T}$ is the label corresponding to the input sample. Thus we have
(6)$$\beta =H^{†}T.$$
here $H^{†}=H^{T}{\left(HH^{T}\right)}^{-1}$ represents the Moore–Penrose Generalized Inverse of the hidden layer output matrix H. To improve the generalization ability of the algorithm, the regularization parameter ***C*** is introduced in (Equation 6), and the output weight matrix can be solved by the following
(7)$$\beta =H^{T}{\left(\frac{I}{C}+HH^{T}\right)}^{-1}T.$$

Then, the output of the ELM can be expressed as
(8)$$f\left(x\right)=h\left(x\right)H^{T}{\left(\frac{I}{C}+HH^{T}\right)}^{-1}T.$$
where h(x) is the output of the previous hidden layer and the input of the ELM.

During the experiments, due to the low accuracy of single ELM, we stack multiple ELMs into Deep ELM to fully exploit fault features in low-dimensional sparse signals. Meanwhile, the kernel function [34] has been applied to many problems due to its strong nonlinear mapping ability, and its most representative application is the Support Vector Machine (SVM) [35]. The kernel function can overcome the curse of dimensionality, and the samples that are linearly inseparable in the original space can be non-linearly mapped to a higher-dimensional space to make them linearly separable, thus improving the classification accuracy. Based on the superior performance of the kernel function, Huang [36] introduced the kernel function into ELM and proposed the KELM algorithm to further enhance the generalization ability and stability. In this paper, the kernel function and DELM are combined to construct a Deep Kernel Extreme Learning Machine (DKELM). The structure is shown in Figure 3.

The original input data of DKELM are abstracted by *k* hidden layers to the input feature $X^{k}$, and then the kernel function maps the feature $X^{k}$. Then, there is no need to give its specific form for the output matrix of KELM; just use the inner product principle of the kernel function to calculate the expression form of $HH^{T}$ and $h\left(x\right)H^{T}$. $HH^{T}$ can be expressed by the kernel function K(x,y) as follows
(9)$$\left\{\begin{array}{c} \Omega_{ELM}=HH^{T}\\ \Omega_{i,j}=h\left(x_{i}\right)\cdot h\left(x_{j}\right)=K\left(x_{i},x_{j}\right) \end{array}\right.$$

Since $h\left(x\right)H^{T}$ can be expressed as
(10)$$h\left(x\right)H^{T}=\left[\begin{array}{c} K\left(x,x_{1}\right)\\ \cdots \\ K\left(x,x_{N}\right) \end{array}\right]$$

The output of DKELM should be
(11)$$f\left(x\right)=\left[\begin{array}{c} K\left(x,x_{1}\right)\\ \cdots \\ K\left(x,x_{N}\right) \end{array}\right]{\left(\frac{I}{C}+\Omega_{ELM}^{T}\right)}^{-1}T.$$

This is equivalent to using a kernel function to replace the mapping of the hidden nodes. The above-mentioned kernel function satisfies the Mercer theorem [37], so this paper adopts a widely used Radial Basis Function (RBF) kernel [38] form
(12)$$K\left(x,x_{i}\right)=\exp\left(-\frac{{∥x-x_{i}∥}^{2}}{\gamma }\right).$$
where γ is the nuclear parameter. The specific form of the feature mapping function h(x) of the hidden layer node in DKELM does not need to be specifically given; the specific form of the kernel function K(x,x) can be used to find the value of the output function. Based on these abstract features rather than the original sample data, the kernel function calculation was used to replace the inner product operation in the high-dimensional space; we then mapped the feature to the higher-dimensional space for accurate classification.

### 2.4. Training

This module trains the DKELM network with a layer-by-layer greedy method and saves the ELM-AE output weights obtained by the least square method. The input weights of each hidden layer are initialized with the output weights of ELM-AE, and layered unsupervised training is performed. DKELM does not require a reverse fine-tuning process, so it is characterized by having a fast speed. The idea of DKELM is to minimize the reconstruction error so that the output can be infinitely close to the original input. After training by each layer, it can learn more advanced features of the original data. Figure 3 describes the training process of the DKELM module. The input sample *X* is used as the target output of the first ELM-AE $X_{1}=X$. Then, the output weight $\beta_{1}$ of the first ELM-AE is used as the input weight $W_{1}$ of the first hidden layer. The output matrix $H_{1}$ of the first hidden layer is used as the input and target output ($H_{1}=X_{1}=X$) of the next ELM-AE, and so on, for training layer by layer. The last hidden layer is trained with KELM, $H_{i}$ is the output matrix of the last hidden layer, and (Equation 11) is the output weight of DKELM. Because of the application of the kernel function, data features that were originally low-dimensional linearly inseparable can be mapped to high-dimensional linearly separable space, making the classification more accurate. Lastly, the PSO algorithm was used to optimize the number of hidden layer nodes, the regularization coefficient, the penalty coefficient and kernel parameters of KELM.

## 3. Experimental Results and Analysis

### 3.1. Experimental Configuration

We use ALINX’s FPGA development board as an industrial embedded platform, which is equipped with Xilinx Zynq UltraScale + MPSOC XCZU9EG, Quad Cortes-A53 1.5GHz industrial-grade chips. The industrial embedded platform is equipped with a Linux system. DDR4 is 8 GB, eMMC is 32 GB and the maximum computing power can reach 3.6TOPS. The actual development board is shown in Figure 4a, which is suitable for algorithm debugging in the laboratory and can be deployed in industrial scenarios by improving heat dissipation and reinforcement, as shown in Figure 4b. We implement our method and all its variants using Python3.8 and Pytorch version 1.7.0 with CUDA 10.2 and CUDNN 7.0.

### 3.2. Compressed Ratios

Compared with traditional Nyquist sampling, CS can compress the original signal more thoroughly. Through random unequal interval sampling, the sampling frequency can be far less than the Nyquist requirement. This will greatly reduce the amount of sampling data, thereby reducing the data storage space and the number of calculations. As the main feature information, the non-zero value of the transform domain will not decrease. Although many incoherent interference values will be generated, we can extract the non-zero value from the transform domain through the iteration of “Threshold detection–Calculating larger interference–Eliminating interference–Lowering the threshold”. Thereby, the original signal can be restored with high fidelity. As shown in Figure 5, the fault signal of the outer race is taken as an example for reconstruction (The amplitude of outer race changes greatly, so the reconstruction is the most difficult among all circumstances). From top to bottom are the comparison curves between the original signal and the reconstructed signal with compressed ratios of 0.9, 0.8 and 0.5. The orthogonal basis of compressed sampling is the Discrete Cosine Transform (DCT) basis, and the measurement matrix is a Gaussian random matrix. The essence of signal reconstruction is to solve the minimum value of the L1 norm.

To evaluate the error of the reconstructed signal more accurately, this paper uses the RMSE as the error measurement index
(13)$$RMSE=\sqrt{\frac{1}{N}\sum_{n=1}^{N}{\left(x\left(n\right)-\hat{x}\left(n\right)\right)}^{2}}.$$
where x(n) is the original signal, and $\hat{x}\left(n\right)$ is the reconstructed signal. Figure 6 shows the RMSE curve of four fault conditions under different compressed ratios.

When the CR is 0.5, the reconstructed signal can completely restore the waveform of the original signal’s fault shock characteristics, and the RMSE at this time is mainly caused by the noise contained in the original signal. When the CR is 0.8, the reconstructed signal can still better reflect the mainshock characteristics, and the signal is slightly distorted. When the CR is 0.9, the reconstructed signal is severely distorted, and the fault shock characteristics cannot be extracted. Combined with the RMSE curve of the reconstructed signal, for the bearing vibration signal used in this paper, as long as the CR is kept below 0.8, the shock characteristic waveform in the original vibration signal can be accurately reconstructed. This means that when the CR is 0.8, the compressed sampling signal still contains all the fault information, which can be directly used for fault diagnosis.

### 3.3. Accuracy and Efficiency of CS-DKELM

The use of the CS-DKELM method can greatly improve the accuracy and efficiency of fault diagnosis. To verify the effectiveness of the fast diagnosis method proposed in this paper, 300 test samples randomly selected in the early stage were input into the CS-DKELM module to test the accuracy and real-time performance of fault diagnosis.

The classification result of the test set is shown in Figure 7. The actual output result is completely consistent with the expected output result (the data label), and the fault classification accuracy can reach 100%. To ensure the reliability of the experimental results, the average value of 20 experiments was adopted. A comparison with the method of directly identifying time-domain signals [39] and the other two methods before the improvement was added, and the results are shown in Table 2.

Compared with the methods DELM and DKELM, which directly identify the time-domain signal, our method performs feature identification on the frequency-domain compressed sampling signal of the original signal. Retaining only 20% of the original sampling points, our method can achieve a diagnosis time an order of magnitude faster. Compared with CS-DELM, which only uses compressed sampling, our method has better adaptability to sparse signals, can ensure higher diagnostic accuracy and achieve good generalization. Currently, the real-time requirements of most industrial systems are under 100 ms. This method can meet the real-time requirements of current industrial systems well while maintaining ultra-high accuracy. It is worth mentioning that the CS-DKELM method is not only for a certain scenario of fault diagnosis but a methodological system [40] of fast fault diagnosis in industrial embedded systems under data-driven conditions. For fault classification problems in different scenarios, we only need to adjust the number of hidden layers and the compressed rate to make it adapt better.

### 3.4. Effectiveness Analysis of Key Factors

#### 3.4.1. The Effect of Hidden Layer Numbers

In the process of training an effective machine learning model, parameter optimization has always been the focus issue for researchers. Considering the effect of hidden layer numbers on the diagnosis accuracy and generalization ability when training the DKELM network, this paper takes simulation experiments to determine the best hidden layer numbers based on the highest fault diagnosis accuracy. We construct DKELM networks with 2, 3, 4 and 5 layers, respectively, and analyze the fault data with the CS-DKELM method. Figure 8 shows the effect of the hidden layer numbers in the DKELM network on the accuracy of fault diagnosis.

It can be seen that the accuracy of DKELM with 2 hidden layers stabilized at around 97% after 6 iterations of optimization. The accuracy of DKELM with 3 hidden layers can reach more than 99% after 6 iterations of optimization. The accuracy of DKELM with 4 hidden layers stabilized at around 87% after 5 iterations of optimization. The DKELM with 5 hidden layers has the lowest accuracy. The number of PSO iterations was 10, and the experimental results are shown in Figure 9. The diagnosis accuracy of the DKELM network with 3 hidden layers can reach more than 99%, and the standard deviation does not exceed 0.44. When the number of hidden layers is greater than 4, the fault diagnosis accuracy begins to decrease significantly, which proves that the DKELM network with 3 hidden layers has the best performance in this scenario. Therefore, we choose the DKELM network with 3 hidden layers to form the CS-DKELM method for the research of this paper.

#### 3.4.2. The Effect of Compressed Ratio

The size of the measurement matrix is closely related to the compressed ratio and the original signal dimensions. Therefore, the compressed ratio can be used to control the size of the measurement matrix, thereby controlling the compressed degree of the sampled signal. Generally, the higher the compressed ratio is, the fewer sampling points obtained by compressed sensing would be, then the higher the efficiency would be; because fewer sampling points means smaller sample memory and smaller data processing time. However, there is an upper limit on the compressed ratio. Too high a compressed rate chosen will cause the random matrix to be unable to obtain the complete information of the original signal; that is, the obtained compressed sampling signal will have serious information loss. This section studies the changes in fault diagnosis accuracy and diagnosis time between 50% and 95% of the compressed ratio. The results are shown in Figure 10.

It can be seen from Figure 8 that the gradual increase in the compressed ratio will reduce the diagnosis time, but the accuracy will also gradually decrease. When the compressed ratio reaches 80%, the accuracy ratio decreases insignificantly, and the diagnosis time is greatly reduced. It shows that the accuracy and efficiency of real-time detection can be guaranteed at the same time. Therefore, by weighing the diagnosis accuracy and the diagnosis time, this paper selects a compressed ratio of 80%. It also shows that when the sampling frequency is 1/5 of the Nyquist sampling, the compressed sensing method could restore the original signal with high fidelity. The direct use of compressed sampling signals can efficiently implement fault diagnosis while reducing storage costs and diagnosis time. In addition, we found that when the compressed ratio reaches 95%, there is still a high diagnostic accuracy. Therefore, for some time-sensitive scenarios where the diagnosis accuracy required is relatively loose, a high compressed ratio can be used, which will greatly reduce the amount of sampling data. Thereby alleviating the pressure of data storage and communication, eventually greatly reducing the time for fault diagnosis.

#### 3.4.3. The Effect of PSO

This paper utilizes an 80% compressed ratio and 3 hidden layers to set up comparative experiments. The diagnosis accuracy and diagnosis time were compared between CS-DKELM and the optimization method CS-DKELM(PSO). The number of hidden layer nodes of the CS-DKELM was manually set to [100,50,100], and the optimization interval of the number of hidden layer nodes of CS-DKELM(PSO) was set to [10:1000]. The results of diagnostic accuracy and diagnosis time are shown in Table 3.

According to the analysis of Table 3, the diagnostic accuracy of the CS-DKELM(PSO) is higher than that of the CS-DKELM, but the diagnosis time is slightly increased. Through the analysis of the PSO algorithm, it is known that the diagnosis time is mainly affected by the number of particle swarm iterations. By drawing the fitness curve, we find that the number of particle swarm iterations for the optimal solution is two times. In this paper, the initial number of iterations of the particle swarm was set to 10, so some additional iteration times are added. By setting the maximum number of iterations to jump out of the loop, the time added by the optimization algorithm can be almost ignored. The parameters of CS-DKELM optimized by PSO are shown in Table 4. The parameter selection principle of CS-DKELM is as simple as possible, and the accuracy rate is more than 90%.

### 3.5. Comparison with Other Methods

To verify the effectiveness of the method in this paper, two classic methods (SVM [41] and DBN [42]) and three latest methods (SPBO-SDAE [11], PSO-DNN [12] and CS-IMSNs [13]) were selected for comparison. The brief settings of these fault diagnosis methods are listed in Table 5.

The three evaluation indicators of average diagnosis time, average diagnosis accuracy and ouput data dimension of the above six methods are respectively compared. The experimental results of the above various methods are shown in Table 6.

It can be seen from Table 6 that due to the compressed sensing used in the method of this paper, the input data dimension is much smaller than that of other methods, which will greatly reduce the fault diagnosis time. Compared with the two classic algorithms, SVM and DBN, our method has greatly improved both the fault diagnosis time and the diagnosis accuracy. The improved algorithm SPBO-SDAE has a certain similarity with our method, but we have higher accuracy and stronger stability. The optimization algorithm PSO-DNN uses a deep neural network. Although the accuracy has been improved, its diagnosis time is obviously insufficient (the training time is too long) compared with our method. The most obvious advantage of our method is that the amount of sample data needed is small, which is very important for reducing storage costs and diagnosis time. Finally, for the CS-IMSNs algorithm, which also uses compressed sensing, due to the need to train deep neural networks of multiple scales, there is a significant gap in training time compared with the method proposed in this paper. Although the fault diagnosis accuracy of the CS-IMSNs algorithm can reach almost 100% and maintain a high degree of stability, it is difficult to achieve for resource-constrained industrial embedded systems. Therefore, our method is an optimal choice for industrial IoT scenarios.

### 3.6. Improvements for Real-Time Data Flow

The improvement of our method for real-time data flow is also worth mentioning. Suppose that the real-time data flow of window size *W* needs to be fault detected within the time interval *T* and W>1/T. There will be (W−1/T) data points to be recalculated at time *t*. If the calculation results of the previous moment can be used, only incremental calculations need to be performed on the new data, which will greatly improve the efficiency of fault diagnosis. As shown in Figure 11 below, the red diamond area is the repeated calculation part.

In the compressed sampling module, any $y=<y_{t},y_{T}>$ is very friendly to incremental calculation, and it is only necessary to input $x_{T}$ into (Equation 1) to obtain the incremental compressed sampling signal. The compressed sampling signal at the current moment can be obtained by splicing the incremental compressed sampling signal with the result of the previous moment $x_{t-T}$. This process can reduce the fault detection time by up to 20%. We compared the improved method with the original method and the specific results are shown in Figure 12. It can be seen that with the increase of the sliding window size and the diagnosis frequency, the diagnosis time for real-time data flow will increase significantly, but our improved method can achieve a speedup of about 20%.

### 3.7. Validation on Actual Hardware Platform

The PT006 series rotating machinery vibration analysis and fault diagnosis experimental platform XZZZ-1 made by VALENIAN company was used as the verification platform. The platform is composed of a data acquisition system, a variable speed drive motor, a frequency conversion controller, a gearbox, a magnetic powder brake, a tension controller and a control cabinet, as shown in Figure 13.

In each experiment, the speed of the variable speed drive motor was set to 1450 r/min, and the sampling frequency was 48 kHz. To better simulate the actual fault situation, we obtained two sets of bearing inner race faults, two sets of bearing outer race faults and two sets of ball faults by replacing the prefabricated defective bearings. Moreover, two sets of gear faults were obtained by replacing the prefabricated defective gears, and one set of the shaft imbalance fault was obtained by adjusting the balance weight of the rotating disc on the shaft. Some faults, such as fatigue, plastic deformation and wear, were real natural faults. Other faults, such as electrical discharge machining (EDM), electric engraver and broken teeth, were man-made faults. Including the normal state, a total of 10 different health conditions were prefabricated on the above-mentioned platform, as shown in Table 7. The data preprocessing method is the same as this paper. The sliding window intercepts non-overlapping vibration signals. Each sample contains 4800 sampling points, 200 samples are intercepted for each health condition. A total of 70% of the samples were randomly selected as the training set, while the remaining 30% were the test set.

We used the CS-DKELM method to perform a fast fault diagnosis on the sampled data, and the average value of 20 trials is shown in Table 8. It can be seen that the average classification accuracy of our method can reach more than 99% with good stability. The diagnosis time still meets the real-time requirements of industrial embedded systems. Compared with other baseline methods, the biggest advantage of our method is that its diagnosis speed is very fast. This near-real-time fault diagnosis method has great practical significance for monitoring equipment health status and preventing equipment damage in industrial IoT scenarios. Our method can be directly applied to industrial embedded platforms as an effective algorithm for mechanical equipment health monitoring. Thus far, the effectiveness of this method in the fault diagnosis of rotating machinery has been verified.

## 4. Conclusions

In this work, we propose an integrated method for data sampling and fast fault diagnosis based on CS and DKELM. By compressed sampling the original monitoring signal, CS-DKELM can achieve substantial compression of monitoring data while preserving fault information. This sampling method for mechanical vibration signals greatly improves the efficiency of subsequent fault diagnosis. In the fault diagnosis stage, through the multi-layer nonlinear learning of the low-dimensional sampling values of the original signal, the adaptive extraction of fault features and the fast diagnosis of fault types are realized. We verified the accuracy and efficiency of the CS-DKELM method, and its influencing factors were studied and compared with existing methods. The analysis shows that the intelligent diagnosis method proposed in this paper has higher real-time performance and diagnostic accuracy in the resource-constrained industrial embedded systems, which are superior to the existing methods. Moreover, only a small amount of monitoring data needed to be sampled in our scheme, which greatly reduces the pressure of transmission, storage and calculation in the process of fault diagnosis. Finally, we applied this method to the actual hardware platform and achieved ideal experimental results, which verified the practical application value of the method proposed in this paper. In the future work, we will combine edge intelligence technology to study anomaly detection technology for multi-source time series data in industrial embedded platforms.

## Figures and Tables

**Figure 1 sensors-22-03997-f001:**
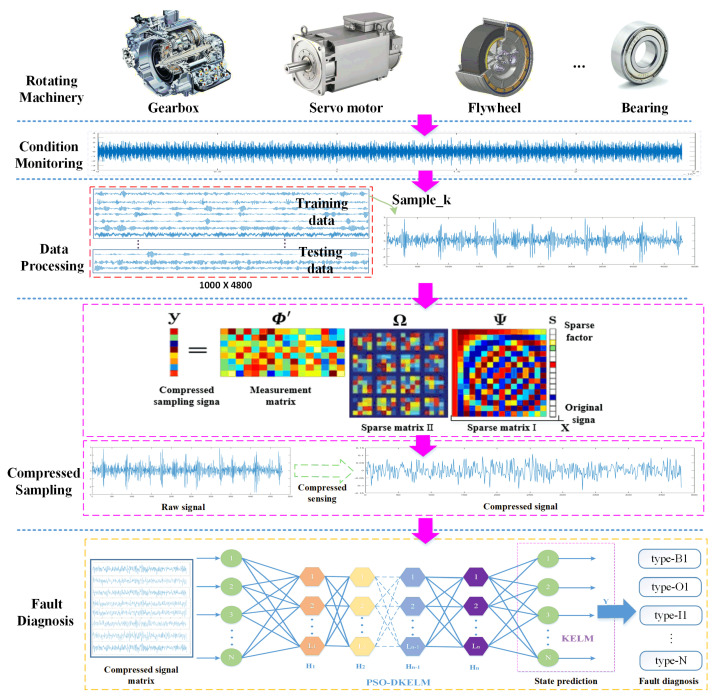
The procedure of the proposed CS-DKELM method.

**Figure 2 sensors-22-03997-f002:**
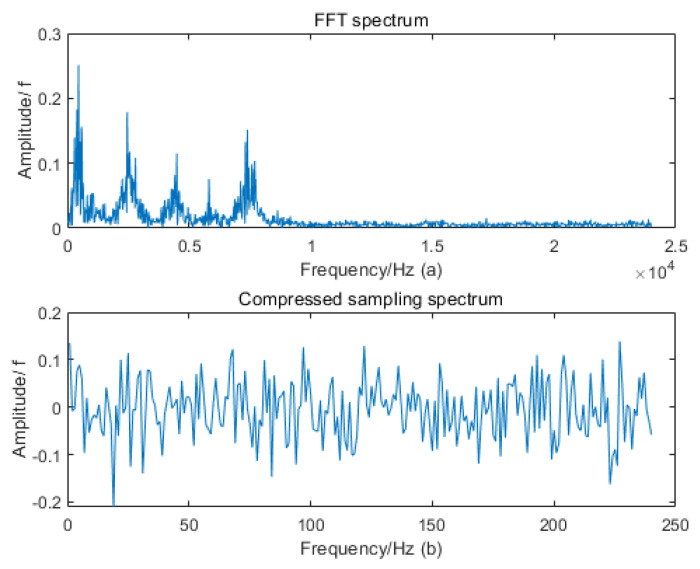
Feature representation of compressed sampling signals in the transform domain. (**a**) Spectrogram with Fourier transform matrix added to the original signal. (**b**) The compressed sampling signal of spectrogram.

**Figure 3 sensors-22-03997-f003:**
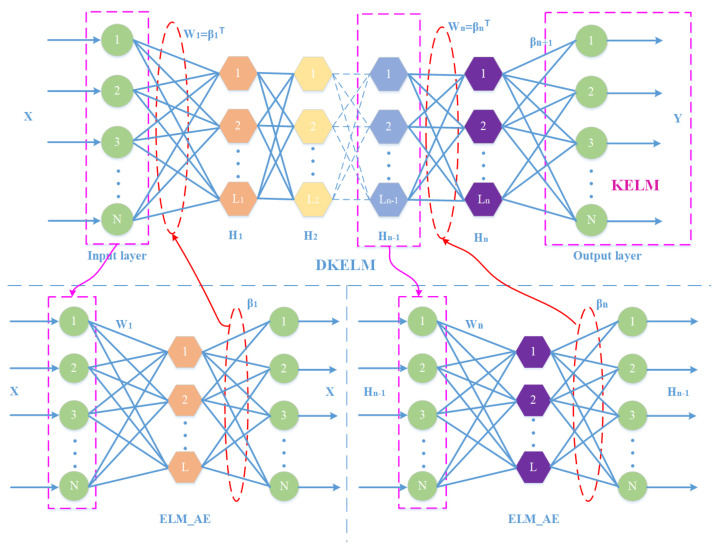
Structure diagram of DKELM.

**Figure 4 sensors-22-03997-f004:**
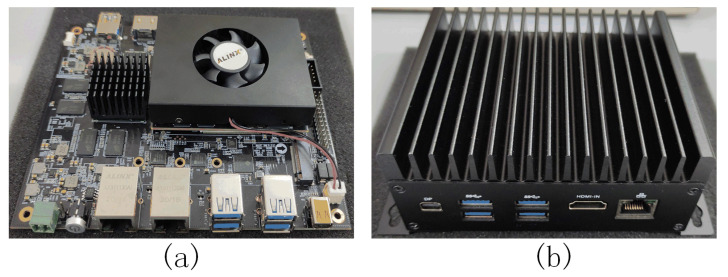
Industrial embedded platform. (**a**) is a industrial embedded platform for debugging, (**b**) is a industrial embedded platform for industrial deployment.

**Figure 5 sensors-22-03997-f005:**
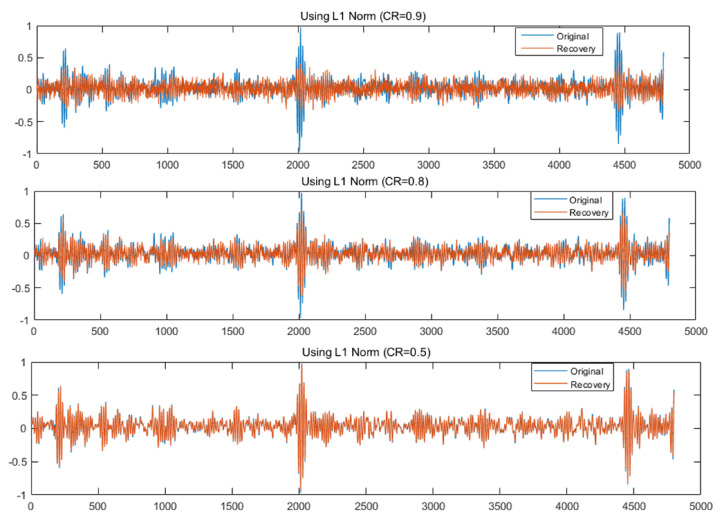
Reconstructed waveform of outer race fault.

**Figure 6 sensors-22-03997-f006:**
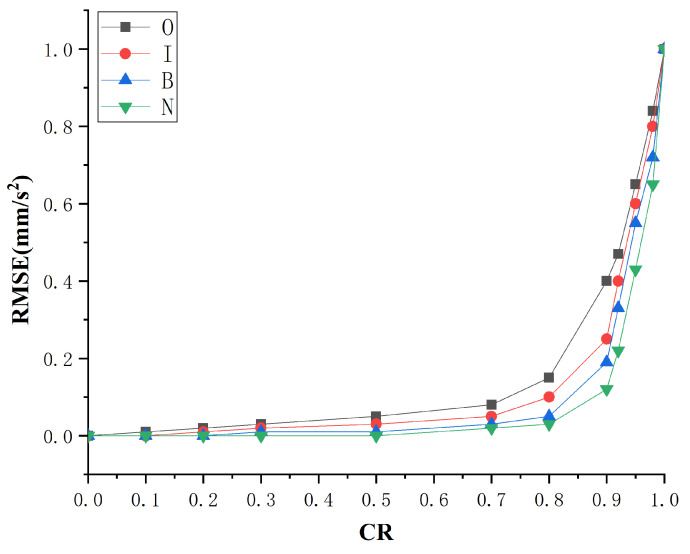
The RMSE curve of the reconstructed signal.

**Figure 7 sensors-22-03997-f007:**
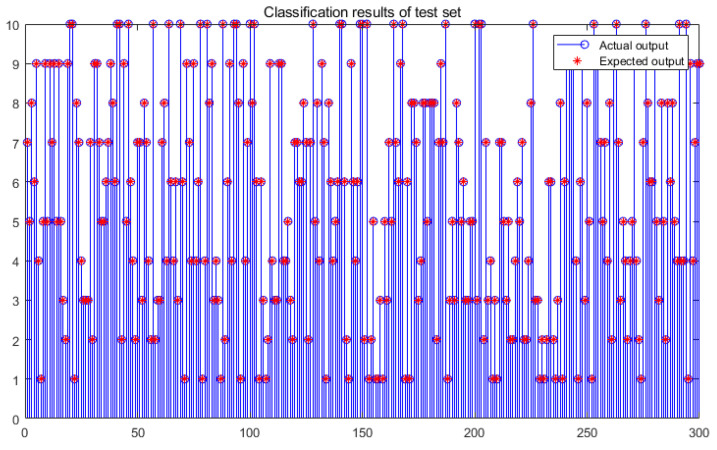
Fault classification result of test set.

**Figure 8 sensors-22-03997-f008:**
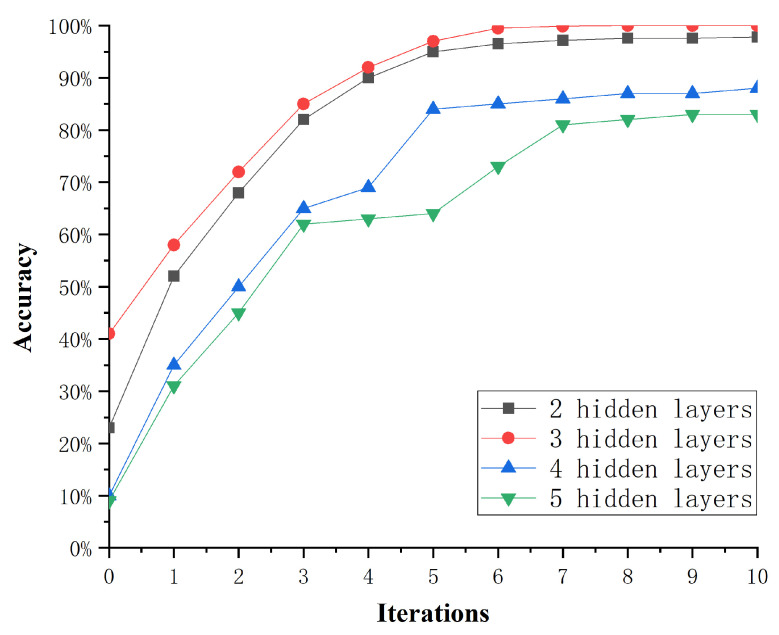
The effect of the hidden layer numbers.

**Figure 9 sensors-22-03997-f009:**
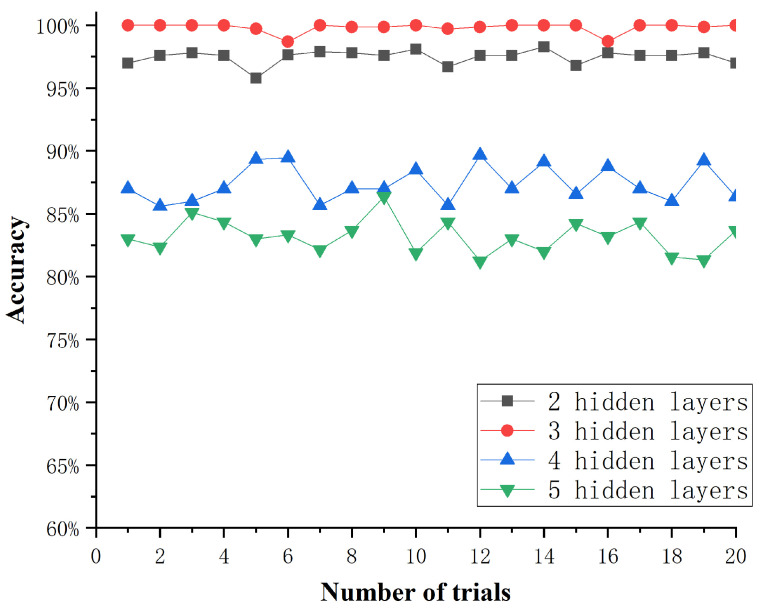
Accuracy of 20 simulation experiments.

**Figure 10 sensors-22-03997-f010:**
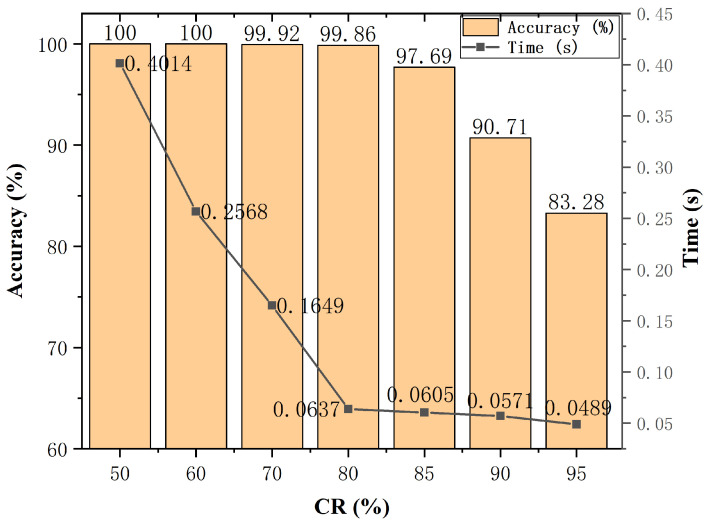
Diagnosis accuracy and time under different compressed ratios.

**Figure 11 sensors-22-03997-f011:**
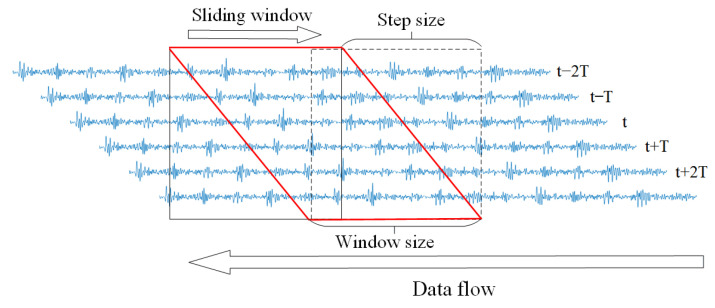
Schematic diagram of the repeated calculation part of the real-time data flow.

**Figure 12 sensors-22-03997-f012:**
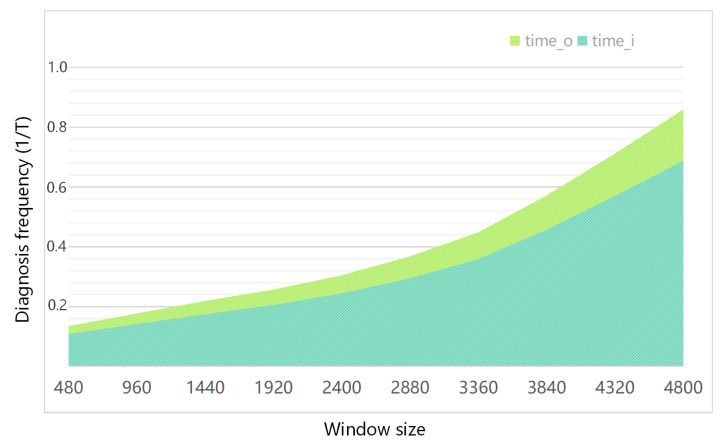
Comparison of time overhead before and after incremental calculation optimization.

**Figure 13 sensors-22-03997-f013:**
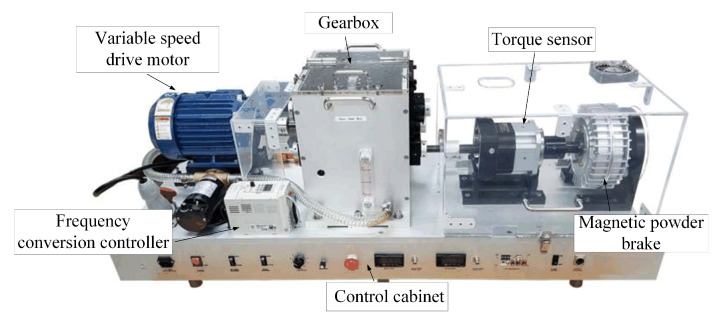
Rotating machinery vibration analysis and fault diagnosis experimental platform XZZZ-1.

**Table 1 sensors-22-03997-t001:** Bearing Data Set $D/D^{\prime }$.

Data Set	Dimension	Number	Fault Type	Diameter (mm)	Label
$D/D^{\prime }$	4800/960	100	B	0.07	1
		100	B	0.14	2
		100	B	0.21	3
		100	O	0.07	4
		100	O	0.14	5
		100	O	0.21	6
		100	I	0.07	7
		100	I	0.14	8
		100	I	0.21	9
		100	N	0	10

**Table 2 sensors-22-03997-t002:** Comparison of our method before and after improvement.

Method	Testing Accuracy (%)	Accuracy Standard Deviation (%)	Testing Time (s)	Time Standard Deviation (%)
DELM	94.67	2.54	0.2109	0.0873
DKELM	97.36	1.78	0.2904	0.0313
CS-DELM	90.71	4.17	0.0589	0.1495
CS-DKELM	99.97	0.44	0.0607	0.0007

**Table 3 sensors-22-03997-t003:** The effect of PSO on diagnosis accuracy and time.

Method	Testing Accuracy (%)	Accuracy Standard Deviation (%)	Testing Time (s)	Time Standard Deviation(%)
CS-DKELM	97.31	0.72	0.0598	0.0023
CS-DKELM (PSO)	99.97	0.44	0.0607	0.0007

**Table 4 sensors-22-03997-t004:** Comparison of parameters.

Method	Nodes of Hidden Layers	Regularization Coefficient	Penalty Coefficient	Kernel Parameters
CS-DKELM	100-50-100	100	100	100
CS-DKELM (PSO)	65-57-94	111.9482	583.2509	210.2382

**Table 5 sensors-22-03997-t005:** Parameter settings for different fault diagnosis methods.

Types	Settings
CS-DKELM (PSO)	Hidden layers: 65-57-94, Kernel: Gaussian radial basis function, Compressed rate: 0.2, Number of particles: 5
SVM	Kernel: Gaussian radial basis function, Box constraint: 1, γ: 1/1024
DBN	Epoch: 20, Batch size: 64, Learning rate: 0.0002, Optimizer: Adam, Momentum: 0.5
SPBO-SDAE	Hidden layers: 46-98-32, Sparse coefficients: 0.2497-0.3216-0.1415, Input data zero ratio: 0.05, Number of particles: 30, Maximum iterations: 500
PSO-DNN	Hidden layers: 600-200-100, Epoch: 50, Learning rate: 0.05, Optimizer: tanh, Momentum: 0.05
CS-IMSNs	Compressed rate: 0.25, Measurement matrices: Gaussian, Epoch: 10, Batch size: 64, Learning rate: 0.0002, Decay: 0.9, Optimizer: RMSProp, Optimizer: ReLU

**Table 6 sensors-22-03997-t006:** Comparison of output results of different methods.

Method	Average Time (s)	Average Accuracy (%)	Ouput Data Dimension
CS-DKELM (PSO)	0.16 ± 0.23	99.97 ± 0.44	1000 × 960
SVM	10.47 ± 2.79	90.66 ± 1.52	1000 × 4800
DBN	13.69 ± 3.27	96.34 ± 3.85	1000 × 2400
SPBO-SDAE	29.31 ± 2.38	98.24 ± 2.36	1000 × 4800
PSO-DNN	147.22 ± 4.26	98.40 ± 0.87	1000 × 4800
CS-IMSNs	270.47 ± 8.93	99.61 ± 0.14	1000 × 1200

The input data dimensions are unified as 1000 × 4800.

**Table 7 sensors-22-03997-t007:** Health status data on the experimental platform.

Dimension	Label	Health Condition
4800	1	Bearing inner race fault (Fatigue: pitting)
	2	Bearing inner race fault (Electric engraver)
	3	Bearing outer race fault (Fatigue: pitting)
	4	Bearing outer race fault (Plastic deform: indentations)
	5	Bearing ball fault (EDM)
	6	Bearing ball fault (Electric engraver)
	7	Gear tooth broken
	8	Gear tooth surface wear
	9	Shaft misalignment
	10	Normal

**Table 8 sensors-22-03997-t008:** Performance of CS-DKELM on the actual hardware platform.

Method	Average Accuracy (%)	Accuracy Standard Deviation (%)	Average Time (s)	Time Standard Deviation(%)
CS-DKELM (PSO)	99.67	0.47	0.17	0.23
SVM	85.22	4.52	7.65	3.52
DBN	90.23	5.85	9.42	3.85
SPBO-SDAE	92.74	3.36	19.83	2.36
PSO-DNN	96.81	2.87	68.92	4.27
CS-IMSNs	97.40	0.40	190.43	8.90

## Data Availability

The datasets used in this study are public datasets released by the Case Western Reserve University Bearing Data Center, which can be obtained from the official website https://engineering.case.edu/bearingdatacenter (accessed on 25 April 2022).

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
