# Peer review of "Fast Fault Diagnosis in Industrial Embedded Systems Based on Compressed Sensing and Deep Kernel Extreme Learning Machines"

_sensors, 2022, doi:10.3390/s22113997_

Round 1

Reviewer 1 Report

This paper presents a data-driven general method for fast fault diagnosis inspired by the ideas of compressed sensing (CS) and deep extreme learning machines (DELM). It is an interesting paper. However, the authors need to address the following minor issues before this manuscript is accepted:

  1. As given in Figure 1, the rotating components are discussed. However, some useful vibration diagnosis methods for detecting the faults in the rotating components were not discussed, such as given in ‘A statistical feature investigation of the spalling propagation assessment for a ball bearing’ and ‘Dynamic modelling of the defect extension and appearance in a cylindrical roller bearing’. The above materials and references can be discussed too.
  2. The specific fault types used for the presented method should be discussed in Section 2. The fault types can be marked in Figure 1.
  3. A key issue is that the test case are stable conditions. However, in the industry cases, the working conditions are very complex, which should not be the stable ones. Can the proposed method be used for the practical industry cases?
  4. Moreover, some experiential values were used in the proposed method. Some brief descriptions for the chosen reasons should be discussed too.

Reviewer 2 Report

This paper developed a data-driven general method for fast fault diagnosis. And two strategies were applied, that are, data sampling and fast diagnosis. Overall, this paper is well structured and promising results are presented. Here are some comments for further improve the quality of this paper:

1) Please check the statement that 'which are optimized from thress aspects:... and fault classification resespectively'. Based on my understanding, data sampling is different from feature extraction and fault classification, and it is unreasonable to put these three items together. Please explain more.

2) In the literature review. the authors stated that most of the existing research on fault diagnosis has achieved vaulabe results. However, no review on the fast fault diagnosis was presented. A short review on the existing research is suggested to included, such as cyclstationary analysis for bearing and gear fault diagnosis (10.1109/TII.2022.3169465). Also, some other techniques such as kurtosis and RMS for fault diagsis are suggested to briefly included.

3) In Fig. 2, the FFT spectrum is presented. It is suggested to include the unit for y-axis.

4) The y-axis of Fig. 10 is with bad visuality.

5) PSO is used in this paper. Why does not choose GA. More explanations should be given.

6) A schematic diagram of the setup in Fig.13 is suggested to be included.

7) Some recommendations for future work should be included in the Conclusion.

Reviewer 3 Report

In this work, the authors propose a new integrated method for data sampling and fast fault diagnosis based on compressed sensing (CS) and deep extreme learning machines with kernel function (DKELM). In the same time the particle swarm optimization algorithm (PSO) is used to optimize the nodes of each hidden layer and to achieve the best fault classification results. According to the authors this method for mechanical vibration signals greatly improves the efficiency of subsequent fault diagnosis. The method was implemented on experimental results in order to be validated. Very good results were obtained, being underlined a faster diagnosis time.

The paper presents in detail the current state of the art in the field, citing numerous works, mostly recent. The elements of originality are also concretely emphasized.

Small corrections would be needed in English to solve some fluency issues.

The tables are not quoted consecutively in the text (for example, table 5 on page 9, line 243). To correct "Table 5. This is a table caption." (page 14) and to specify what the data in table 5 represent.

The paper includes many abbreviations, not all explained.

Round 2

Reviewer 2 Report

This paper has been improved by addressing reviewers' comments. I have no extra comments. I agree to accept it.